# Fluorination of Terminal Groups Promoting Electron Transfer in Small Molecular Acceptors of Bulk Heterojunction Films

**DOI:** 10.3390/molecules27249037

**Published:** 2022-12-18

**Authors:** Tao Chen, Rui Shi, Ruohua Gui, Haixia Hu, Wenqing Zhang, Kangning Zhang, Bin Cui, Hang Yin, Kun Gao, Jianqiang Liu

**Affiliations:** 1School of Physics, Shandong University, Jinan 250100, China; 2School of Physics and Electrical Engineering, Kashgar University, Kashgar 844000, China

**Keywords:** fluorination, organic photovoltaic, bulk heterojunction, stacking, electron transfer

## Abstract

The fluorination strategy is one of the most efficient and popular molecular modification methods to develop new materials for organic photovoltaic (OPV) cells. For OPV materials, it is a broad agreement that fluorination can reduce the energy level and change the morphology of active layers. To explore the effect of fluorination on small molecule acceptors, we selected two non-fullerene acceptors (NFA) based bulk heterojunction (BHJ) films, involving PM6:Y6 and PM6:Y5 as model systems. The electron mobilities of the PM6:Y5 and PM6:Y6 BHJ films are 5.76 × 10^−7^ cm^2^V^−1^s^−1^ and 5.02 × 10^−5^ cm^2^V^−1^s^−1^ from the space-charge-limited current (SCLC) measurements. Through molecular dynamics (MD) simulation, it is observed that halogen bonds can be formed between Y6 dimers, which can provide external channels for electron carrier transfer. Meanwhile, the “A-to-A” type J-aggregates are more likely to be generated between Y6 molecules, and the π–π stacking can be also enhanced, thus increasing the charge transfer rate and electron mobility between Y6 molecules.

## 1. Introduction

In recent years, organic photovoltaic (OPV) cells have developed rapidly and the photoelectric conversion efficiency (PCE) of OPV cells has been over 19% [1,2,3,4]. Designing new organic photoactive materials, including electron donor and acceptor materials, is one of the most effective strategies for improving the PCE of OPV cells. In the last decades, great efforts have been devoted to developing molecular design methods to modulate the optoelectronic properties and intermolecular aggregation of materials [5,6,7,8]. Among these, introducing electron-withdrawing halogens, such as fluorine, chlorine, and bromine, into the conjugated building blocks, has been demonstrated to be one of the most efficient strategies for OPV materials [9,10].

Fluorination is one of the most effective molecular modification methods to adjust the photoelectric properties of π-conjugated materials [11]. For OPV materials, the introduction of fluorine atoms can effectively reduce the energy level and adjust the morphology [12]. Recently, some high-performance OPV devices based on fluorine atoms have been reported in succession [13,14]. Zou and coworkers synthesized the new fluorine-containing non-fullerene acceptor (NFA) material Y6 and mixed it with the fluorine-containing donor material PM6 as the active layer of OPV devices, yielding a high PCE of 15.7% [15]. These results show that the fluorination strategy is an important method to optimize the performance of OPV devices. However, although fluorinated donor or acceptor materials have great potential in the preparation of high-efficiency OPV devices, few studies have been reported on the impact of fluorination strategies on the charge transport performance of OPV materials, especially in OPV systems containing fluorination. Therefore, it is essential to explore the effect of fluorination on the charge transport performance of OPV materials.

In this work, we selected PM6:Y5 and PM6:Y6 bulk heterojunction (BHJ) films to investigate the effect of end-group fluorination of small molecule acceptor materials on charge transport properties. To quantify the charge transport properties, we extracted the charge carrier mobilities and energetic disorder values of the PM6:Y5 and PM6:Y6 systems by using the space-charge-limited current (SCLC) and the Gaussian disorder model (GDM) measurements. The PM6:Y6 system exhibits high mobility and low energetic disorder of electron transport. To explore the origin of electron mobility differences, molecular dynamics (MD) was adopted to simulate the molecular stacking of PM6:Y5 and PM6:Y6 systems. The results show that the F atoms in the Y6 molecules can form halogen bonds with the F, N, and O atoms in the adjacent Y6 molecules, and the formed halogen bonds enhance the channels for intermolecular electron transfer. Fluorination of terminal groups makes it easier for A–D–A type NFA molecules to form the “A-to-A” type J-aggregates, and the π–π stacking is strengthened. This work explains the effect of fluorination on the charge transport behavior of small molecule acceptors and enriches insights into charge carrier transport in OPV materials.

## 2. Results and Discussion

### 2.1. Optical Properties

Figure 1a shows the chemical structures of PM6, Y5, and Y6 molecules. The energy band diagram of materials is shown in Figure 1b. It can be observed that fluorine-containing materials exhibit the down-shifted lowest unoccupied molecular orbital (LUMO) and highest occupied molecular orbital (HOMO) energy levels relative to the donors and acceptors without fluorination due to the strong ability to regulate the π-electron behaviors for fluorination [16,17,18]. The absorption spectra normalized to polymer donor and NFAs acceptor is shown in Figure 1d. The neat Y6 film presents a significant red-shift of 35 nm relative to Y5, which can be attributed to the stronger π–π stacking interaction and highly ordered aggregation in the Y6 film due to the strong hydrogen bonding of fluorine atoms in Y6 [19]. The broad absorption spectra of Y6 in a long wavelength range further improve their ability to harvest incident photons. Appendix A show the steady-state photoluminescence (PL) of the PM6, Y5, Y6, and their blend films and time-resolved photoluminescence (TRPL) spectra of the neat Y5, neat Y6, and PM6:Y5 and PM6:Y6 blend films, respectively. The neat PM6, Y5, and Y6 films exhibit clear PL emission with emission peaks at approximately 695 nm, 890 nm, and 906 nm, respectively. In the PM6:Y5 and PM6:Y6 blend films, the emission peak of PM6 at 695 nm is almost invisible, indicating that efficient charge transfer occurred between PM6 and Y5(Y6). As shown in Appendix A, the pure Y6 and PM6:Y6 blend films have shorter exciton lifetimes, which means that excitons can dissociate faster in the pure Y6 and PM6:Y6 blend films. Figure 1c shows the JV curves of OPV devices with a conventional architecture of ITO/PEDOT:PSS/BHJ/PDIN/Al under the standard 1-Sun illumination, where PEDOT:PSS and PDIN are used as hole and electron transport layers, respectively. The corresponding device parameters are summarized in Table 1. The PM6:Y6 devices delivered an average PCE of 15.87% with an average open-circuit voltage (V_OC_) of 0.828 V, an average short-circuit current density (J_SC_) of 25.12 mA/cm^−2^, and an average fill factor (FF) of 76.31%. Compared to the Y6-based cells, the PM6:Y5 devices exhibit a lower average PCE of 6.83%, with a lower average J_SC_ of 12.72 mA/cm^−2^, an average FF of 57.47%, and an average V_OC_ of 0.934 V. It is worth noting that the larger J_SC_ value of PM6:Y6-based devices can be partly attributed to the broadened and enhanced photo response of the Y6 acceptors.

### 2.2. Charge Transport Properties and Morphology Analysis

The fluorination strategy will reduce the energy levels of OPV materials, considering that the energy level alignment of PM6 and Y5 is not rational. We study the charge transport properties of PM6:Y5, PM6:Y6, PBDB-T:Y5, and PBDB-T:Y6 BHJ films. The space-charge-limited current (SCLC) measurements are performed to evaluate their electron and hole transport properties. The zero-field electron mobility of the organic film can be expressed as:(1)$$J=\frac{9}{8}\varepsilon_{0}\varepsilon_{r}\mu_{0}exp\left(0.89\beta \sqrt{\frac{V}{d}}\right)\frac{V^{2}}{d^{3}}$$
where J is the space-charge-limited current density, V is the space-charge-limited voltage, d is the thickness of the active layer, $\varepsilon_{0}$ is the dielectric constant of vacuum, $\varepsilon_{r}$ is the relative dielectric constant of the polymers, $\mu_{0}$ is the zero-field mobility, and β is the Poole–Frankel (PF) slope. Figure 2a and Appendix A display the electron and hole carrier transport results for the PBDB-T:Y5, PBDB-T:Y6, PM6:Y5, and PM6:Y6 BHJ films. The PM6:Y6 BHJ films have the highest electron and hole mobilities, while the PM6:Y5 BHJ films have the lowest electron and hole mobilities. Under the same donor condition, the Y6 systems tend to have higher electron and hole mobilities, which means fluorinated acceptors can be helpful for charge carriers’ transport in BHJ films.

To further evaluate the electron transport properties in BHJ films, we investigate the temperature-dependent electron mobilities in electron-only devices (PBDB-T:Y5, PBDB-T:Y6, PM6:Y6, and PM6:Y5) to extract *σ_e_*. In organic semiconductors, the energetic disorder *σ* describes the width of the state distribution in energy [20]. Energetic disorder *σ_e_* represents the standard derivation of the Gaussian distribution of the lowest unoccupied molecular orbital levels. GDM is employed to extract the energetic disorder *σ_e_*, which is expressed as:(2)$$\mu_{0}=\mu_{\infty }exp\left[-{\left(\frac{2\sigma_{e}}{3k_{B}T}\right)}^{2}\right]$$
where *μ*_0_ is zero-field mobility, *μ_∞_* is high-temperature limit mobility, *k_B_* is the Boltzmann constant, and T is the temperature [21,22]. As shown in Figure 2b, the *σ_e_* values of PM6:Y6 and PM6:Y5 are 47 eV and 59 eV, respectively. The results suggest that the PM6:Y6 BHJ films have less energetic defects and low energetic disorder, which can be beneficial for improving the device performance and enhancing the dissociation of the charge transfer state [23].

The 2D Grazing-Incidence Wide-Angle X-ray Scattering (GIWAXS) patterns and corresponding 1D profiles in the in-plane (IP) and out-of-plane (OOP) directions are shown in Figure 3. The PM6:Y6 blend film show an obvious (010) peak in the OOP direction but no (010) signal in the IP direction, indicating that the donor and acceptor molecules prefer the face-on dominated π–π stacking [24]. The π–π stacking distance (d) of the (010) peaks of the blend film was calculated quantitatively. As shown in Table 2, The d-spacings of PM6:Y5 and PM6:Y6 films are 3.57 Å and 3.55 Å, respectively. Compared with PM6:Y5 films, PM6:Y6 films show higher peak (010) diffraction intensity, indicating that there is a stronger π–π stacking interaction. Further quantitative calculations show that the crystal coherence length (CCL) of PM6:Y6 films is larger, which indicates that PM6:Y6 films are more tightly stacked. Charge transport and exciton dissociation are more effective. As shown in Appendix A, the height image of the PM6:Y5 blend film shows an Rq value of 1.165 nm. In contrast, PM6:Y6 shows an increased Rq of 2.623 nm, indicating weakened donor–acceptor miscibility. This means that the introduction of F atoms could boost acceptor aggregation.

### 2.3. Molecular Dynamics Simulation

The hopping model is used to calculate the electron mobility of organic semiconductors [25]. The charge transfer rate between adjacent molecules can be evaluated using the semi-classical Marcus theory [26]:(3)$$k_{ij}=\frac{V_{ij}}{h}\sqrt{\frac{\pi }{\lambda k_{B}T}}exp\left[-\frac{{\left(∆G_{ij}+\lambda \right)}^{2}}{4\lambda k_{B}T}\right]$$
where h is the reduced Planck’s constant, *V_ij_* is the transfer integral between the initial and final states, *k_B_* is the Boltzmann constant, T is the temperature and is set to 300 K, *λ* is the reorganization energy, which is defined as the energy change associated with the geometry relaxation during the charge transfer, and $∆G_{ij}$ is the relevant change of total Gibbs free energy. Considering that electrons are transported only between acceptor molecules, $∆G_{ij}$ equals zero and Equation (3) then becomes:(4)$$k_{ij}=\frac{V_{ij}}{h}\sqrt{\frac{\pi }{\lambda k_{B}T}}exp\left[-\frac{\lambda }{4k_{B}T}\right]$$

From the above equation, we can observe that the electron transfer rate of organic semiconductors mainly depends on the recombination energy and the transfer integral. The reorganization energy has both internal and external contributions, with the internal contribution arising from changes in the geometry of the molecular dimer when the electron transfer takes place, and the external contribution arising from changes in the surrounding media that accompany the charge transfer [27]. In organic crystals, the external contribution is usually ignored and only the internal contribution needs to be considered. *λ* is estimated to be 138 meV for Y5 and 143 meV for Y6 using DFT calculations at the B3LYP/6-31G(d) level. According to the Marcus rate formula [Equation (4)], small internal reorganization energy and large intermolecular transfer integrals are helpful to speed up the charge transfer processes between neighboring molecules. In the BHJ film, electron carriers mainly transport in acceptor materials, and hole carriers typically transport donor materials [28,29]. Therefore, we calculated the electron transfer integrals between acceptor molecules. Through screening, more Y6 dimer structures have electron transfer integrals (*V_ij_* > 0.1 meV). This means that there are more electron transfer channels between Y6 molecules. The average electron transfer integral is 7.59 meV for Y5 dimers and 8.47 meV for Y6 dimers. Therefore, the results show that Y6 acceptor molecules have a faster electron transfer rate.

The halogen bond assists the intermolecular electron transfer, resulting in the intermolecular electron transfer integral of Y6 being greater than that of Y5. In organic transport materials, a weak interaction is generally considered a way to realize charge transfer between molecules [30]. Halogen bonding, a kind of non-covalent intermolecular interaction, has been revealed by experiments that it can induce charge transfer [31,32]. Therefore, the non-covalent interaction (NCI) has been calculated to quantify the halogen non-covalent molecular interactions by using Multiwfn [33,34]. Figure 4a shows the plot of the reduced density gradient versus the electron density multiplied by the sign of the second Hessian eigenvalue for Y6 evaluated by B3LYP/6-31G(d). The isosurfaces in the blue color lying in F1 and F2 can be found for the Y6 dimer (Figure 4b), indicating a strong intermolecular interaction.

Figure 5 shows the typical stacking modes of the dimer structures of Y5 and Y6 molecules. It can be found that there are significant differences between the molecular stacking of Y5 and Y6. The molecular stacking of Y5 is only π–π stacking. Due to the presence of fluorine in the molecule, Y6 shows two types of intermolecular non-covalent interactions in addition to π–π stacking, involving linear C-F⋯F and C-F⋯N binding modes, and nonlinear C-F⋯F and C-F⋯O (Figure 5). The radial distribution function (RDF) between acceptor molecules is shown in Appendix A. At the same time, the “A-to-A” type J-aggregates also exist in PM6: Y6 (Appendix A) [35]. The results show that fluorinated acceptor materials are more likely to form the “A-to-A” type J-aggregates and the π–π stacking can be enhanced, which is consistent with the results from the GIWAXS measurements. All of these non-covalent interactions contribute to the formation of a three-dimensional stack structure, which can provide a transport channel for charge carriers. As a consequence, the PM6:Y6 system achieves a higher electron transfer integral.

## 3. Materials and Methods

### 3.1. Materials

PM6, PBDB-T, Y5, and Y6 were purchased from Solarmer Materials Inc. (Beijing, China). All the materials were used as received.

### 3.2. Computational Methods

The aggregation and intermolecular interactions of donor and acceptor molecules in the donor–acceptor nanostructured blending film have been explored by a combination of molecular dynamics (MD) simulations and density functional theory (DFT) calculations. All atomistic MD simulations are performed using the GROMACS 5.1.1. software [36]. The atom types and intra-/inter-molecular interaction parameters of PM6, Y5, and Y6 are built from the general AMBER force field (GAFF) with the RESP charges [37,38]. A spherical cut-off of 1.3 nm is used for the summation of van der Waals interactions and short-range Coulomb interactions. A Particle–Mesh Ewald solver is used for long-range Coulombic interactions. The simulations are carried out with 3D periodic boundary conditions using the Verlet integrator with a time step of 1.0 fs.

The PM6:Y5 and PM6:Y6 BHJ films are built and imitated using the following procedure (Appendix A): (1) the initial model of PM6:Y6 (Y5) complex with 1:1 weight ratio is built by random placing 20 PM6 with 4 repeat units and Y6 (Y5) into the periodic box at dimensions of 150 Å × 150 Å × 150 Å using the Packmol program (seeding); (2) 1 ns of simulation to minimize system energy (minimization); (3) 1 ns of simulation at 300 K and 100 bar to make molecules close together (compression); (4) 5 ns of simulation at 600 K and 1 bar, then cooling down to 300 K in 3 ns (thermal annealing); And (5) 10 ns of equilibration at 300 K and 1 bar (equilibration). The velocity-rescaling thermostat [39] and the Berendsen barostat [40] under the NPT pattern are applied to control the temperature and pressure, respectively.

### 3.3. Fabrication of the Electron-Only Device

The electron-only devices were fabricated with a configuration of ITO/Al(50 nm)/BHJ/PDIN/Al (Appendix A). The ITO substrates with a sheet resistance of 15 Ω square-1 were cleaned in deionized water, acetone, ethanol, and isopropanol with ultrasonic treatment for 20 min. These ITO substrates were then treated with UV-ozone for 15 min to improve their work function. An Al layer (50 nm) was deposited on the ITO substrates via evaporation in a vacuum. Subsequently, PM6:Y6 (1:1 wt%) was dissolved by heating and stirring at 40 °C in chlorobenzene (12 h). Then, the dissolved solution (24 mg/mL) was spin-coated onto the ITO substrates in a nitrogen-filled glove box, and the spin-coated active layer was annealed at 100 °C for 10 min. After the annellation, a PDIN methanol solution (2 mg/mL) with 0.3 vol% of acetic acid was a spin coating on the top of BHJ. An Al layer (100 nm) was deposited on the PDIN layer via evaporation in a vacuum. The fabrication procedure of the hole-only device and OPV cells was similar to the electron-only device. The J-V curves were measured by a source measure unit Keithley 2612b.

## 4. Conclusions

In summary, we investigated the effect of terminal group fluorination on electron transport in OPV acceptor materials of BHJ films by using both experimental and MD simulation methods. From the SCLC measurement, it can be observed that the PM6:Y6 system has higher electron mobility than the PM6:Y5 film. Through the MD simulation, we found that the “A-to-A” type J aggregates are more likely to generate between Y6 molecules in PM6:Y6 BHJ film, which enhances the π–π stacking. In addition to π–π stacking, there are non-covalent interactions, such as C-F⋯F and C-F⋯N halogen bonds. These non-covalent interactions can provide more electron transfer channels between acceptors. This work shows that the fluorination of terminal groups can change the stacking pattern and provides support for the rationality of the fluorination strategy for OPV materials.

## Figures and Tables

**Figure 1 molecules-27-09037-f001:**
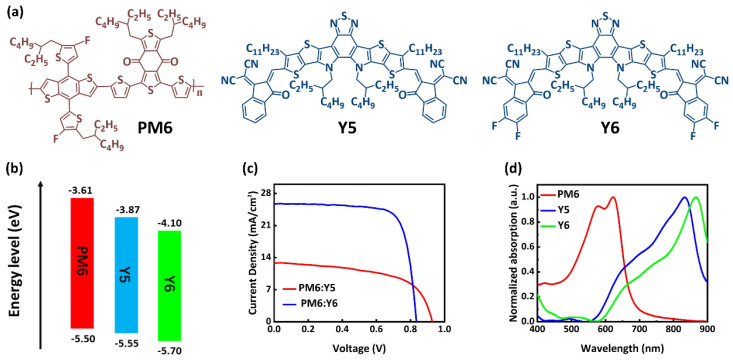
(**a**) Chemical structures and (**b**) energy level diagrams of donors (PM6) and acceptors (Y5 and Y6); (**c**) J−V characteristics of the PM6:Y5 and PM6:Y6 BHJ solar cells under 1−Sun illumination; and (**d**) normalized absorption spectra of PM6, Y5m and Y6.

**Figure 2 molecules-27-09037-f002:**
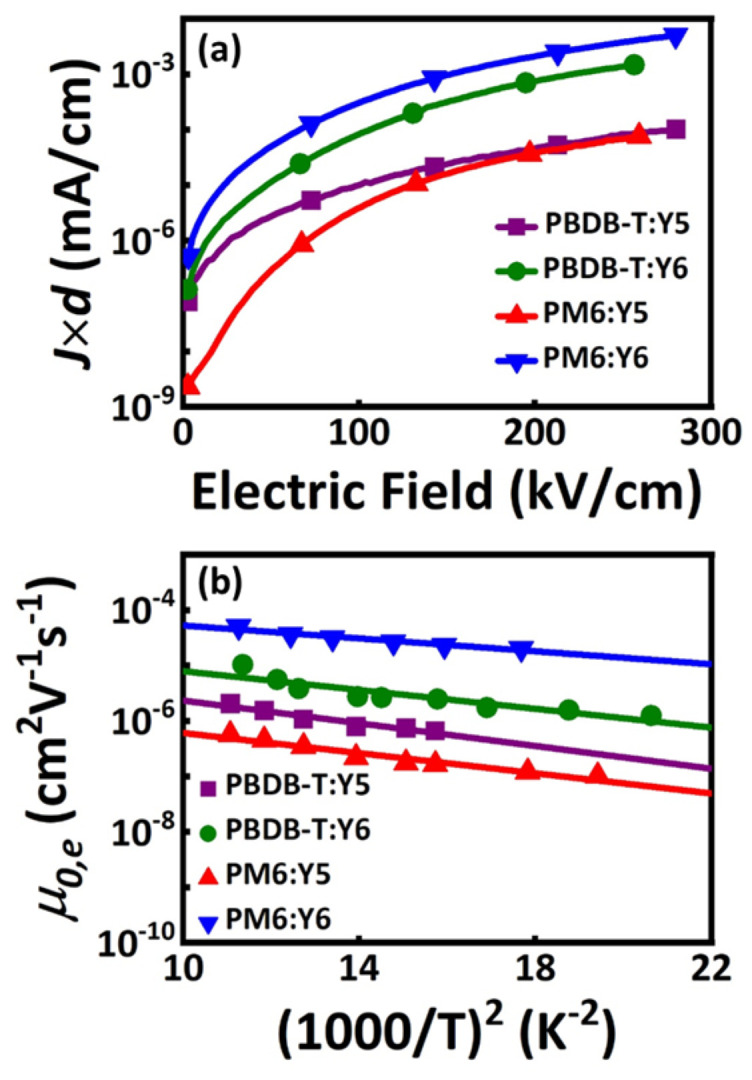
Electron carrier transport results for the PBDB−T:Y5, PBDB−T:Y6, PM6:Y5, and PM6:Y6 BHJ films. (**a**) *J* × *d* as a function of the applied electric field; and (**b**) zero−field mobility as a function of (1000/T)^2^.

**Figure 3 molecules-27-09037-f003:**
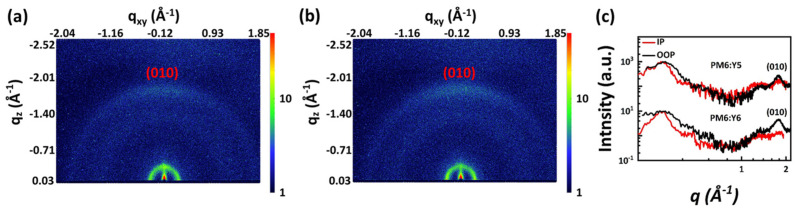
Two−dimensional GIWAXS patterns of the (**a**) PM6:Y5 and (**b**) PM6:Y6 blend films and (**c**) the diagram of scattering intensity for the PM6:Y5 and PM6:Y6 blend films.

**Figure 4 molecules-27-09037-f004:**
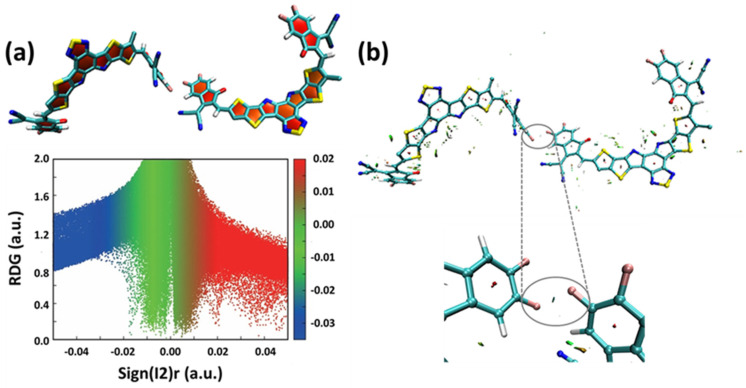
(**a**) Plot of the reduced density gradient versus the electron density multiplied by the sign of the second Hessian eigenvalue for Y6 evaluated by B3LYP/6−31G(d). (**b**) The NCI diagram for the Y6 dimer.

**Figure 5 molecules-27-09037-f005:**
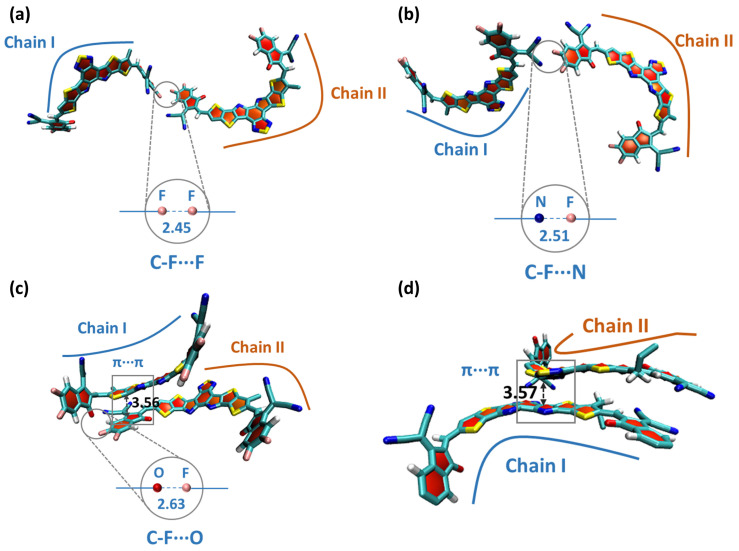
Representative configurations of the Y6 dimer (**a**) containing C−F⋯F halogen bonding, (**b**) C−F⋯N halogen bonding, (**c**) C−F⋯O halogen bonding and π⋯π stacking in Y6 neat films; and (**d**) the Y5 dimer only involving π⋯π stacking in Y5 neat films.

**Table 1 molecules-27-09037-t001:** The detailed photovoltaic parameters of the OPV cells.

Active Layer	V_OC_ (V)	J_SC_ (mA cm^−2^)	FF (%)	PCE (%)
PM6:Y5	0.934 ± 0.006	12.72 ± 0.20	57.47 ± 1.39	6.83 ± 0.16
PM6:Y6	0.828 ± 0.005	25.12 ± 0.29	76.31 ± 0.68	15.87 ± 0.22

**Table 2 molecules-27-09037-t002:** Details of parameters of the (010) peak for the GIWAXS measurement.

Active Layer	Location (Å^−1^)	d (Å)	FWHM (Å^−1^)	CCL (Å)
PM6:Y5	1.76	3.57	0.27	20.94
PM6:Y6	1.77	3.55	0.25	22.62

## Data Availability

Data will be provided upon request.

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
