# Peer review of "Fluorination of Terminal Groups Promoting Electron Transfer in Small Molecular Acceptors of Bulk Heterojunction Films"

_molecules, 2022, doi:10.3390/molecules27249037_

Round 1

Reviewer 1 Report

The manuscript by C. Tao et al. is devoted to the important field of development of functional materials for energetics. The authors report investigation of the influence of chemical modification of novel non-fullerene acceptors on the properties of OPV cells.

The article makes a good overall impression. The manuscript is well written. The results presented are supported by the data reported both in main article and supporting information.

I would like to suggest some minor changes in the manuscript to improve its quality.

1. In my opinion the materials and methods section should be placed before the results and discussion.

2. I recommend to add in the main text references and corresponding discussions for all data presented in the supporting information. In the current version there are no references to the SI in the text presented that rise a lot of questions on the first read.

Reviewer 2 Report

The authors of this manuscript investigated the effect of end groups halogenation on electron transport in OPV acceptor materials of BHJ films by using both experimental and MD simulation methods. The manuscript is generally well written but is still peppered with grammar and typographic errors and requires a thorough edit.

1. Page 3. line 77NFAs acceptor is shown in Figure 1cshould be Figure 1d

2. Page 3. line 81Figure 1d showsshould be Figure 1c

3. Page 4. Figure 2 should be marked (a) and (b)

4. The figure should be in the same format, for example, figure 4 and figure 5.

5. In the supporting information, The relevant information of the Figure S2, Figure S3 and Figure S4 are not found in the manuscript. What these experiments can show is not mentioned in the manuscript.

Reviewer 3 Report

The authors reported the fluorinated-NFA acceptor effect on the device performances using PM6:Y5 and PM6:Y6 as examples, and they used SCLC measurements and MD simulations to investigate the distinctness between the PM6:Y5- and PM6:Y6-blend films. The detailed comments are listed below:

1. The authors used halogenation as the title of the manuscript, but they only took Y5 and Y6 as examples to demonstrate their concepts. I suggest the authors change their title from halogenation to fluorination otherwise they should add the chlorinated-NFA acceptor to the manuscript as well.

2. The novelty of this manuscript is not so significant. The factors that influenced the devices’ performance plenty, such as device structure, blend ratio, processing solvent, additives, energy offsets, energy loss, and so on. It seems the molecular packing way is the key factor that causes the huge difference in device performance between PM6:Y5- and PM6:Y6-based blend films. From the device parameters listed in Table 1, I would not say the difference in Voc, Jsc, and FF between PM6:Y5- and PM6:Y6-based devices only in molecular packings of acceptors.

3. What is the blend ratio and the total concentration of PM6:Y5 and PM6:Y6? The device fabrication procedures are missing in the manuscript, only the electron-only device fabrication method was found, and what is the hole-only device structure used in the experiment?

4. The authors added the PBDB-T: Y5 and PBDB-T: Y6 devices into the mobilities tests for comparison, but there is no discussion mentioned in the manuscript. The reasons for adding these devices should be referred and more discussion should be provided.

5. Figures S2 to S6 were not mentioned in the manuscript. If they are not essential to the manuscript, they should be removed otherwise more discussion should be added to the manuscript.

6. The difference between PM6:Y5 and PM6:Y6 in 2D GIWAX patterns was ambiguous. The position of the (010) peak in the OOP direction should be pointed out in the 2D GIWAXS patterns and the GIWAXS profiles. In addition, the calculation method and how to get the d-spacings values of two blend films should be addressed.

7. How many samples were used for the device performance comparison? The standard deviation should be added in Table 1.

8. What is the coherence length value for both PM6:Y5 and PM6:Y6 blend films?

9. A typo error was found in the manuscript. There are two equations 3 but no equation 4.

Round 2

Reviewer 2 Report

After the author's revision, I think that this manuscript has met the requirements for publication without change.

Author Response

We sincerely thank the reviewer for thoroughly examining our manuscript and providing very helpful comments to guide our revision.

Reviewer 3 Report

Although the authors have made some revisions, there are still some ambiguous descriptions in the manuscript. The following are the comments based on my reading:

1. Figure S2 was only mentioned using one sentence without any interpretation, the authors should be responsible for giving the reasons why they conducted the experiments and what they try to explain to the readers in the manuscript.

2. The same problem was found for adding PBDB-T: Y5, and PBDB-T: Y6 blend films into comparison, the reasons should be addressed and described in the manuscript.

3. The meaning of σe values should be explained, the authors need to give some explanation for those values instead of writing down those numbers.

4. What are the CCL values for all blend films? The authors should calculate those values instead of saying the PM6:Y6 films is larger without giving any numbers or experimental evidence in the revised manuscript.

5. The location, d-spacing, FWHM, and CCL values at the 010 peak of both films should be summarized into a table and added to the discussion in the manuscript.

6. From the GIWAXS patterns, it seems there is not too much difference between PM6:Y5 and PM6:Y6 blend films. Why these films showed a huge difference in their mobilities?

7. Why the blend ratio was 1:1 in a simulated PM6:Y6 blend film, but that was 1:1.2 in a real device?

8. Why the Rq values in PM6:Y5 and PM6:Y6 blend films are 6.44 and 8.75 nm in Figure S4, respectively, but those values are 1.165 and 2.623 nm in the text of the manuscript?
